# Exploring the Prevalence and Resistance of *Campylobacter* in Urban Bird Populations

**DOI:** 10.3390/vetsci11050210

**Published:** 2024-05-11

**Authors:** Aida Mencía-Gutiérrez, Francisco Javier García-Peña, Fernando González, Natalia Pastor-Tiburón, Iratxe Pérez-Cobo, María Marín, Bárbara Martín-Maldonado

**Affiliations:** 1Department of Animal Physiology, Faculty of Veterinary Sciences, Complutense University of Madrid, 28040 Madrid, Spain; 2Grupo de Estudio de la Medicina y Conservación de la Fauna Silvestre (GEMAS), 28220 Majadahonda, Spain; 3Grupo de Rehabilitación de la Fauna Autóctona y su Hábitat (GREFA), 28220 Majadahonda, Spain; 4Department of Pharmacology and Toxicology, Faculty of Veterinary Sciences, Complutense University of Madrid, 28040 Madrid, Spain; 5Laboratorio Central de Veterinaria, Ministerio de Agricultura, Pesca y Alimentación, 28110 Algete, Spain; 6Department of Nutrition and Food Science, Faculty of Veterinary Sciences, Complutense University of Madrid, 28040 Madrid, Spain; 7Department of Veterinary Medicine, School of Biomedical and Health Sciences, Universidad Europea de Madrid, 28670 Villaviciosa de Odón, Spain; bmmjimenezvet@gmail.com

**Keywords:** *Campylobacter*, antimicrobial resistance, urban wildlife, landfills, passerines, AMR, zoonoses, One Health, wild birds, wildlife

## Abstract

**Simple Summary:**

Wildlife has been described as a sylvatic reservoir for a multitude of pathogens. The interactions between wild birds, domestic animals, and humans in urban areas are high, so monitoring urban birds is key for the surveillance of zoonosis such as campylobacteriosis. This infection is mainly caused by thermophilic *Campylobacter* spp., and it is the most reported foodborne zoonosis in the European Union. This study aimed to evaluate the prevalence of thermophilic *Campylobacter* isolates and their antimicrobial resistance pattern in urban wild birds. Results showed that 16.8% of birds were positive for *Campylobacter*, with 82.4% of the isolates resistant to at least one antimicrobial. The taxonomic order of individuals, main diet, age, and season of sampling were significant factors associated with *Campylobacter* spp. carriage. Although the prevalence of *Campylobacter* was low, the rate of antimicrobial-resistant isolates is worrying, so similar studies should be included in the antimicrobial resistance surveillance programs.

**Abstract:**

The increasing urbanization of ecosystems has had a significant impact on wildlife over the last few years. Species that find an unlimited supply of food and shelter in urban areas have thrived under human presence. Wild birds have been identified as amplifying hosts and reservoirs of *Campylobacter* worldwide, but the information about its transmission and epidemiology is still limited. This study assessed the prevalence of *Campylobacter* in 137 urban birds admitted at a wildlife rescue center, with 18.8% of individuals showing positive. *C. jejuni* was the most frequent species (82.6%), followed by *C. coli* and *C. lari* (4.3% each). The order Passeriformes (33.3%) showed significant higher presence of *Campylobacter* when compared to orders Columbiformes (0%) and Ciconiiformes (17.6%), as well as in samples collected during the summer season (31.9%), from omnivorous species (36.8%) and young individuals (26.8%). Globally, *Campylobacter* displayed a remarkable resistance to ciprofloxacin (70.6%), tetracycline (64.7%), and nalidixic acid (52.9%). In contrast, resistance to streptomycin was low (5.8%), and all the isolates showed susceptibility to erythromycin and gentamycin. The results underline the importance of urban birds as reservoirs of thermophilic antimicrobial-resistant *Campylobacter* and contribute to enhancing the knowledge of its distribution in urban and peri-urban ecosystems.

## 1. Introduction

The increasing urbanization of ecosystems has had a significant impact on wildlife over the last years. While some species have suffered population decline, many others have adapted and thrived under human presence [1]. Most of these species find in urban areas a supply of unlimited resources for their biological functions, and the increase in urban waste has promoted landfills as an important source of food for urban wildlife [2,3]. In this sense, due to the constant food availability and the milder winter temperatures, some migratory species such as white storks (*Ciconia ciconia*) have shortened or even stopped their migration in the Iberian Peninsula during the last decades [4]. Nevertheless, a diet based on urban waste may involve several risks such as nutritional deficiencies, intoxications, or the acquisition of antimicrobial-resistant (AMR) bacteria, including *E. coli*, *Salmonella*, and *Campylobacter* [5].

*Campylobacter* is the most reported foodborne zoonotic bacteria in the European Union (EU) and has significant socioeconomic repercussions on public health [6,7]. Although approximately 137,000 confirmed cases of human campylobacteriosis have been reported in the European Union in 2022 [7], the actual number could be up to 9 million per year [8]. On the other hand, human campylobacteriosis can lead to reactive arthritis and severe neurological disorders such as Guillain–Barré and Miller–Fisher syndromes [9]. Despite the high host specificity, many animals are considered natural hosts of *Campylobacter*, since the bacteria has been detected as a part of the normal microbiota in cattle, sheep, swine, and poultry, with the highest prevalence in the last one, which is defined as the main reservoir [10,11]. Indeed, it has been reported that more than 88% of human campylobacteriosis cases originate from poultry [12]. In addition, wild birds have also been identified as amplifying hosts and reservoirs worldwide, including Antarctica, with *C. jejuni*, *C. lari*, and *C. coli* as the most reported species [13,14].

Information about the transmission and epidemiology of this genus is still limited, especially in wild bird populations, although evidence about the transmission between wild birds and humans has already been published [6]. Ecological and life history traits, such as feeding habits or sociality, can influence the infection rates of *Campylobacter* in wild birds [13]. Among urban wild birds, Passeriformes and Columbiformes are two orders that have shown a high prevalence of infection [6,9,15,16,17]. In cities, there is a significant level of direct or indirect contact between humans and these species, particularly in parks, playgrounds, market squares, or terraces, which poses a potential risk to public health (Figure 1) [18].

The dissemination of *Campylobacter* by wild birds becomes especially worrying when antimicrobial-resistant (AMR) strains are involved [19,20]. The increasing development and spread of AMR through the environment is concerning, since it is estimated that resistant bacteria are involved in over 5 million deaths [21]. Although the contribution to the development of resistances is not comparable to that due to livestock and human activity [22,23], several studies have reported a high proportion of AMR *Campylobacter* isolates, including multidrug-resistant (MDR) strains, in wild bird populations [22,24,25].

The presence of AMR in wildlife is directly related to the anthropization of ecosystems, but other transmission routes may also exist, such as environment contamination and the dissemination of antimicrobial resistance genes (ARGs) through clouds [26,27]. In urban wildlife, the environmental pressure on antimicrobial residues in cities may be the main route for AMR acquisition, highlighting the importance of including the monitoring of urban wildlife in AMR surveillance. Due to their increasingly close contact with humans, wildlife may act as a reservoir for zoonotic infections [28]. Moreover, *Campylobacter* can easily transfer genetic elements such as ARGs to other bacteria [24].

In this context, this study aimed to assess the prevalence of *Campylobacter* in urban wild birds from different species admitted at a wildlife rescue center, as well as identify strains at the species level, and determine their susceptibility to antimicrobials.

## 2. Materials and Methods

### 2.1. Sample Collection

From 2017 to 2021, 137 birds of 13 species from urban populations admitted to the Grupo de Rehabilitación para la Fauna Autóctona y su Hábitat (GREFA) Wildlife Hospital (Madrid, Spain) were examined and sampled following a standard protocol for the health status monitoring of animals. All birds were handled according to the European Directive 2010/63/EU and the Spanish Royal Decree 53/2013 [29,30]. As a part of the sanitary status analysis, a cloacal swab was aseptically collected from each individual at their arrival to assess the presence of *Campylobacter*, before any treatment. Samples were preserved in ferrous sulfate, sodium metabisulfite, and sodium pyruvate (250 mg/L each) medium (FBP) (Oxoid^®^, Basingstoke, UK) with 0.5% active charcoal (Sigma-Aldrich^®^, St. Louis, MO, USA) and kept frozen at −20 °C until analysis.

Furthermore, the body condition score, age, and gender of all the individuals included in the study were determined. The age of the animals was estimated based on feather development, whereupon they were grouped into young (nestlings and fledglings) or adult categories. Gender was identified by sexual dimorphism when possible. Finally, the visual body composition score (BCS) of each animal was estimated by morphometry using a zero-to-five system, where level 0 represented cachectic birds, 1 emaciated birds, level 2 under-conditioned birds, level 3 well-conditioned birds, level 4 over-conditioned birds, and level 5 obese birds [31].

### 2.2. Campylobacter spp. Isolation and Identification

Isolation of thermophilic *Campylobacter* was performed at the Central Veterinary Laboratory (LCV) of Algete (Ministry of Agriculture, Fisheries and Food) based on the ISO 10272–1:2017 procedures [32], as previously described by Mencía-Gutiérrez et al. [25]. Briefly, samples were streaked onto modified charcoal cefoperazone deoxycholate (mCCDA) and Preston agar (Oxoid^®^, Basingstoke, UK) and incubated in a microaerobic atmosphere at 41.5 ± 1 °C for 44 ± 4 h. The morphology and motility of all the *Campylobacter*-like colonies were assessed under microscopy, and Gram staining was employed for the morphology study (Panreac AppliChem^®^, Darmstadt, Germany). Also, biochemical tests were performed, including catalase activity, oxidase activity (MAST^®^ ID Oxidase Strips, Amiens, France), and hippurate hydrolysis for preliminary *C. jejuni* identification. Moreover, microanaerobiosis and aerobiosis growth were also tested by the culture in blood agar (Oxoid^®^, Basingstoke, UK) at 25 ± 1 °C and 41.5 ± 1 °C for the first one, and at 37 ± 1 °C for the second one.

*Campylobacter* species was confirmed according to the multiplex PCR assay described by Wang et al., including the genes *hipO* and 23 S rRNA, present in *C. jejuni*; *glyA*, present in *C. coli*, *C. lari*, and *C. upsaliensis*; and *sapB2* from *C. fetus* subsp. *fetus* [33]. Isolates with inconclusive results were subjected to a sequential PCR assay described by Denis et al. for simultaneous identification of *C. jejuni* and *C. coli* [34]. Samples with non-determinant results were classified as *Campylobacter* spp. Isolates were cryopreserved at −80 °C in the FBP medium until antimicrobial susceptibility testing.

### 2.3. Antimicrobial Susceptibility Testing

*Campylobacter* strains were subjected to an antimicrobial susceptibility test (AST) using the broth microdilution method. Sensititre *Campylobacter* EUCAMP2^®^ plates (Thermo Fisher Scientific^®^, Madrid, Spain) were used according to the manufacturer’s instructions. Each strain was tested against six different antimicrobials: nalidixic acid (NAL), ciprofloxacin (CIP), erythromycin (ERY), tetracycline (TET), gentamycin (GEN), and streptomycin (STR). The susceptibility or resistance of *C. jejuni* and *C. coli* isolates to antimicrobials was determined using the epidemiological cut-off values (ECOFF) established by the European Committee on Antimicrobial Susceptibility Testing (EUCAST, 2023) (Table 1) [35]. For other *Campylobacter* species, the minimum inhibitory concentration (MIC) cut-off values for *C. jejuni* were applied as they were the most restrictive ones. MDR was defined as resistance to at least three different antimicrobial families [36].

### 2.4. Statistical Analysis

Statistical analysis was performed with two commercially available software packages: SPSS v23.0 (SPSS Inc., Chicago, IL, USA, 2002) and Statgraphics Centurion XVI v16.2.04 (StatPoint Technologies, Inc., Warrenton, VA, USA).

Assuming a binomial distribution for *Campylobacter* shedding and AMR, different statistical tests were performed to assess whether there was an association with different variables. The following variables were selected for statistical analysis: (1) bird order: Ciconiiformes, Columbiformes, Passeriformes; (2) main feeding diet: insectivore, herbivore, and omnivore; (3) presence in landfill: yes or no; (4) season of sampling; and (5) age. Univariate statistical analysis was performed with Pearson chi-square χ^2^ test and Fisher’s exact test when appropriate for independence (*p*-value), using *Campylobacter* spp. status (absence/presence) as the dependent variable. A two-tailed *p*-value < 0.05 was considered to indicate a statistically significant difference. Odds ratio (OR) and 95% confidence intervals (CI_95%_) were also calculated. Logistic regression was also implemented to test the impact of variables with more accuracy and perform a multivariate analysis to assess the correlation between variables simultaneously.

## 3. Results

A total of 137 individual birds from 13 urban avian species and 3 orders (Ciconiiformes, Columbiformes, and Passeriformes) were analyzed for *Campylobacter* carriage. A total of 23 *Campylobacter* isolates were recovered from 8 out of the 13 urban species (61.5%), with an individual prevalence of 16.8% (CI_95%_ 11.4–23.9%). The most frequent species was *C. jejuni* (82.6%), followed by *C. coli* and *C. lari* (4.3% each). Two isolates could not be identified at the species level (8.7%). Further details about the distribution of *Campylobacter* among the different orders and species are shown in Table 2.

Chi-square and Fisher’s exact test showed a significant relationship between four variables and prevalence rates of *Campylobacter*: order, feeding, season of sampling, and age (Table 3). A subsequent univariate logistic regression analysis was performed excluding the *Campylobacter*-free categories to investigate the association between variables with greater accuracy. Overall, the age of individuals and season of sampling resulted in key variables on the *Campylobacter* epidemiology: the proportion of *Campylobacter* was 3.47 times higher in young individuals than in adults (*p* = 0.01) and 8.28 times higher in individuals sampled during summer than in other seasons (*p* = 0.002) (Figure 2). According to the taxonomic order, a tendency was detected in Passeriformes, being 2.33 times more positive to *Campylobacter* spp. than Ciconiiformes (*p* = 0.102). The rest of the variables (diet and presence in landfill) were not significantly associated with *Campylobacter* prevalence. Gender and body condition scores were assessed but discarded as independent variables due to the high number of animals of unknown gender (missing = 103) and the high percentage (90%) of animals classified in only one of the 5 possible categories in the case of body condition.

A multivariate analysis was later performed including the risk factors identified as significant in the univariate logistic regression: season (summer vs. spring) and age (young vs. adult), but also the trend detected for the order in the case of Passeriformes with respect to Ciconiiformes (Table 3). A statistically significant difference was observed in the case of young individuals and samples collected during the summer. Finally, separate logistic regression analyses were performed for adult and young individuals to assess a possible co-linearity with the effect of season. A significant correlation was observed between seasonality and young birds but not in the case of adults.

On the other hand, only 17 *Campylobacter* isolates out of 23 isolates could be subjected to AST; it was not possible to recover 6 isolates (4 *C. jejuni* and 2 *Campylobacter* spp.) from FBP cryovials. Results showed that 82.4% (14/17; CI_95%_: 58.2–94.6%) of all the *Campylobacter* isolates analyzed for AST were resistant to at least one antimicrobial, and only three isolates were pansusceptible (17.6%). Globally, *Campylobacter* displayed a remarkable resistance to CIP (70.6%), TET (64.7%), and NAL (52.9%) and a low resistance to STR (5.8%). All isolates were susceptible to ERY and GEN. The results of AST are summarized in Table 4. The statistical analysis revealed that, among all the species, white storks tended to have higher rates of resistant *Campylobacter* (*p* = 0.052), while not significantly different.

Among the resistant isolates (n = 14), 12 were identified as *C. jejuni*, 1 as *C. coli*, and 1 as *C. lari*. The antimicrobial resistance patterns of *Campylobacter* isolates are summarized in Table 5 and Figure 3. No MDR isolates were detected.

## 4. Discussion

Although some authors consider that the role of wild birds as a reservoir for enteric pathogens may be overestimated [37], others have reported similarities among *Campylobacter* strains isolated from both humans and wildlife [17]. Likewise, several studies have confirmed the presence of the same *Campylobacter* genotypes in wild birds, domestic animals, and humans [6,24,38]. Therefore, the monitoring of wild birds during ringing activities or upon admission to rescue centers would be a very valuable tool for assessing the distribution of these bacteria in the environment.

The presence of *Campylobacter* spp. in fecal samples of urban birds belonging to the orders Ciconiiformes, Columbiformes, and Passeriformes has been previously evaluated, but the individual prevalence found in this study (16.8%) differs from the findings of other authors. Du et al. reported a prevalence of 10.96% in birds from urban and suburban areas of Beijing [24], while Ramonaite et al. detected a prevalence of 36.2% in fresh fecal samples of free-living urban birds in Lithuania [39]. This difference between both studies could be due to two circumstances. First, while the first study included urban wild birds from 33 different species, the second one focused only on two species (crows and pigeons). The second reason could be that Ramonaite et al. included a selective enrichment step to detect stressed thermophilic *Campylobacter* isolates in the samples [39]. Most of the studies about *Campylobacter* prevalence in wild animals often exhibit heterogeneous results among species or taxonomic groups, even within the same study, as observed in this work [40]. On the other hand, the prevalence of *Campylobacter* seems to be linked to the biology and evolution of the avian species, reflecting the evolutionary commensal association between *Campylobacter* and birds [13]. In general, the order Passeriformes exhibited the highest infection rate, being particularly notable in the Corvidae family, as observed by other authors [26,41,42]. The presence of two different *Campylobacter* species (*C. jejuni* and *C. lari*) within this family has also been previously reported [43]. Crows, which commonly have a high prevalence of *Campylobacter* [40,44], are omnivorous and opportunistic scavengers, and it has been described that individuals from peri-urban areas with a greater amount of anthropogenic waste in their stomach contents had a higher prevalence of *Campylobacter* compared to those from rural areas [45]. Similar associations have been observed in white storks with feeding habits in landfills, which have been linked to a higher prevalence and a broader spectrum of *Campylobacter* species [46]. In the Iberian Peninsula, there is a close relationship between resident populations of white storks and landfills, which could explain the higher prevalence observed in this study, compared to the findings of Szczepanska et al. in Poland (7.6%) [46]. However, the present results did not confirm the association between the presence of *Campylobacter* and feeding in landfills. Surprisingly, none of the Columbiformes included in this study was positive for *Campylobacter*, which contrasts with most of the literature [24,38,41,43]. Columbiformes are herbivorous species that may primarily acquire *Campylobacter* infection through contaminated vegetables and surfaces. Specifically, ringdove *(Columba palumbus)* and collared dove *(Streptopelia decaocto)* are birds inhabiting rural areas with lesser interaction with human wastes, which reduces the acquisition of bacteria such as *Campylobacter* [47].

Moreover, differences found among bird species could be related to their body temperature average. In 2022, Casalino et al. reported a strong association between body temperature and *Campylobacter* presence, with a high prevalence in species whose average body temperature ranged from 40.7 to 41.8 °C. Out of this body temperature range, *Campylobacter* was not found [48]. It is well-known that body temperature is higher in small birds [49] such as passerines, which showed the highest prevalence (33.3%). However, the average body temperatures of most of the species included in this study are unknown; therefore, it would be an interesting aspect to evaluate in future studies.

In this study, *Campylobacter* rate infection also seems to be influenced by season and weather conditions. Some studies have reported a higher prevalence of *Campylobacter* in wild birds through the spring/early summer months [25,50], when carriers might increase the pathogen fecal shedding during the moment of acute stress related to the breeding season, infecting in turn their partners or even their offspring. Likewise, a peak of campylobacteriosis cases in humans and domestic animals is often observed during the warmer months of the year, which agrees with the present results [10,38,51]. Also, the age of the animals has been previously addressed as a significant factor in the *Campylobacter* infection rates, thus it was included as a variable in the statistical analysis of this study. Results showed a higher prevalence in young individuals (26.8%) than in adults (9.8%). According to our results, Colles et al. observed that the prevalence of *Campylobacter* was significantly higher in the younger starlings, but while younger starlings showed a greater variety of *Campylobacter* species (*C. jejuni*, *C. coli*, and *C. lari*) compared to adult birds, *C. jejuni* was the most prevalent species in young birds (14/15), and only one isolate of *C. coli* was observed [50]. On the contrary, some studies have found no significant correlation between the age of the birds and the presence of *Campylobacter* spp. [38,52]. Separate logistic regression analyses were performed for adults and chicks to evaluate the effect of the summer season. This was significant only for young birds, indicating a correlation between the two categories, where the highest prevalence of chicks could be attributed to a higher prevalence of *Campylobacter* fecal shedding during the summer season. Although it would be desirable to carry out studies specifically focused on this subject, most research is often conducted as part of ringing programs or time-limited captures for animal welfare reasons. Previous research has also indicated that *C. jejuni* is more frequently detected in younger individuals, while *C. coli* is more common in adults [53]. However, the number of non-*C. jejuni* isolates obtained in the present study (1 *C. coli* in young birds, 1 *C. lari* in adults) were limited, making it impossible to draw any definitive conclusions.

It has been described that gut microbiota composition differs substantially between young and adult birds [54], and although some progress is being made in the wild bird’s microbiota understanding [55], studies regarding the role of bacterial diversity on the presence of *Campylobacter* spp. are still scarce [54]. Nevertheless, previous studies have confirmed that gut microbiota diversity is strongly affected by the urban transformation of the feeding area, and even though the factors that contribute to intraspecies and interspecies variations are still unknown, feeding on human waste could significantly influence the avian gut microbial community [55].

Similarly, diet can influence the infection rate of *Campylobacter* among bird species [14]. Omnivorous and insectivorous birds showed a statistically significant higher prevalence compared to herbivorous birds, which agrees with recent studies [44,48]. Some authors confirmed that omnivorous birds, in particular those that forage on the ground, had higher proportions of *Campylobacter* [44]. Moreover, one of the feeding sources for omnivorous birds is garbage, which could increase the exposure of these animals to *Campylobacter* strains of human origin [3].

The dynamics of AMR in wildlife present a potential hazard to human and animal health, particularly considering that approximately 40% of emerging human diseases are estimated to have originated in wildlife [47]. Over the last decade, clinically relevant antimicrobial-resistant strains, including those from *Campylobacter* spp., have been isolated from various wildlife species [16,41]. In our study, CIP was the antimicrobial with the highest resistance rate, followed by TET and NAL, which agrees with other authors’ findings [13,22,24,44,56]. The combination of these three antimicrobials was the most frequent resistance pattern (CIP-NAL-TET). Fluoroquinolones, such as CIP and NAL, have been commonly used in the treatment of campylobacteriosis, but their effectiveness has decreased due to the rapid selection of CIP-resistant *Campylobacter* isolates [57]. However, CIP remains one of the first-line antimicrobials used in avian medicine, and the spread of CIP-resistant genes can complicate the resolution of infectious diseases. High rates of TET-resistant *Campylobacter* isolates have also been described in wild birds [22]. The transmission of tetracycline resistance seems to be associated with avian reservoirs due to the high average body temperature, which can promote the conjugation of plasmids carrying TET resistance [58]. The increase in fluoroquinolone-resistant and tetracycline-resistant *Campylobacter* isolates in public health has led to the use of other antimicrobial families, such as macrolides, for campylobacteriosis complications, but their effectiveness can be compromised by the increase in AMR strains. Fortunately, neither ERY-resistant nor MDR strains were identified in this work, although some authors have reported MDR rates up to 33% in *Campylobacter* isolates from wild birds [17,24]. The order Ciconiiformes showed the highest rate of AMR, with 91.7% of the isolates tested to AST (12 *C. jejuni* and 1 *C. coli*), showing resistance to at least one antimicrobial agent. In the Iberian Peninsula, white storks commonly rely on landfills as a food source [59] and exhibit altered migration patterns, with many individuals shortening or even stopping migration to Africa. In this sense, it has been suggested that open landfills or wastewater sites could be key for the acquisition and dissemination of AMR among wildlife since they are high-risk environments for the presence of antibiotic residues and antimicrobial-resistant bacteria [3,60,61]. Despite a moderate prevalence in our study and an omnivorous diet with occasional scavenging and presence in landfills, the Corvidae family did not show alarming results concerning AMR.

## 5. Conclusions

The findings of this study underline the relevance of urban birds as reservoirs of antimicrobial-resistant *Campylobacter* strains that could be disseminated in the environment and transmitted to other animals and humans, with significant implications for public health. The acquired data contribute to enhancing the knowledge of the *Campylobacter* distribution in wild birds of urban and peri-urban ecosystems. Our results confirm the seasonality of *Campylobacter* prevalence and highlight the role of taxonomic order, diet, and age of animals as key factors in its distribution among wild birds. Further research should be undertaken on the epidemiology of *Campylobacter* in wild birds.

## Figures and Tables

**Figure 1 vetsci-11-00210-f001:**
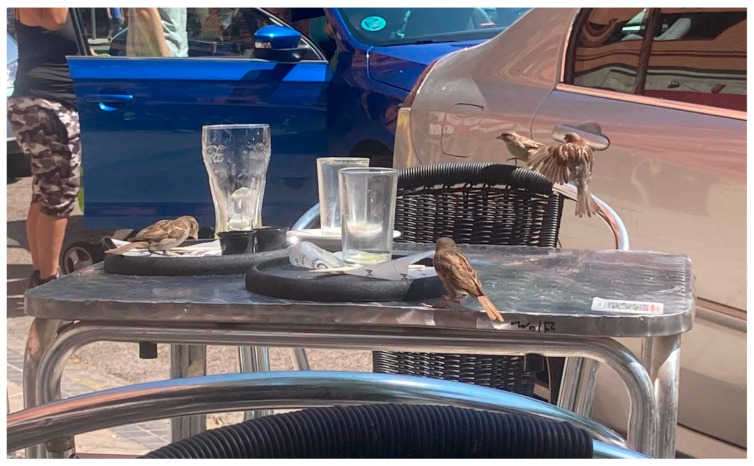
Urban wildlife can pose a risk due to its close contact with humans.

**Figure 2 vetsci-11-00210-f002:**
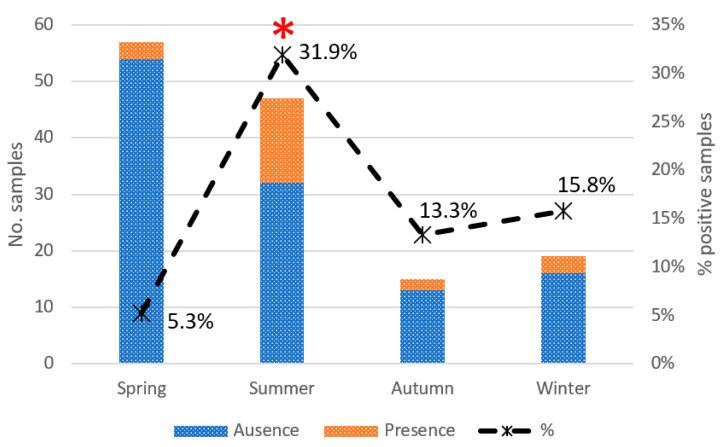
*Campylobacter* prevalence by season in urban birds (* *p* < 0.01). Bars indicate the total number of samples, with different colors depending on the absence (dark) or presence (light) of *Campylobacter*. The dashed line indicates the percentage of positive samples about the total number of samples.

**Figure 3 vetsci-11-00210-f003:**
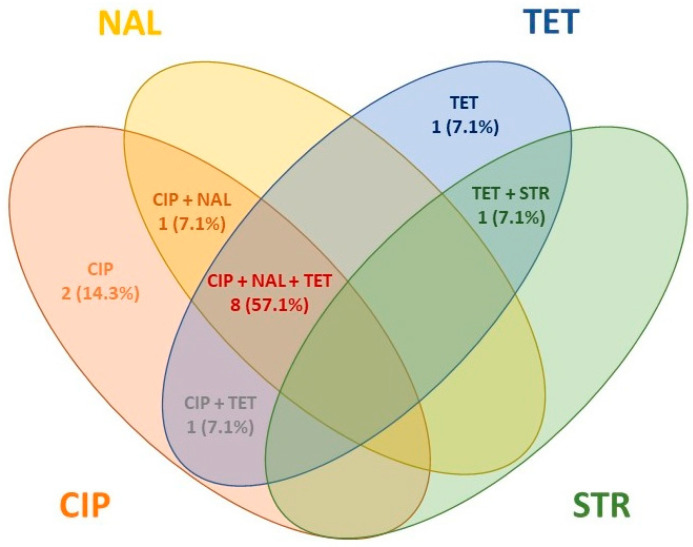
Antimicrobial resistance patterns found in *Campylobacter* isolates.

**Table 1 vetsci-11-00210-t001:** Minimum inhibitory concentration (MIC) cut-off values that were employed for *C. jejuni* and *C. coli* (EUCAST, 2023).

	Cut-Off Values (mg/L)	
	*C. jejuni*	*C. coli*
Nalidixic acid	>16.0	>16.0
Ciprofloxacin	>0.5	>0.5
Erythromycin	>4.0	>8.0
Tetracycline	>1.0	>2.0
Gentamycin	>4.0	>4.0

**Table 2 vetsci-11-00210-t002:** *Campylobacter* prevalence and species regarding urban bird species.

Species *	N	Feeding ^a^	Presence in Landfills ^b^	Positive Birds	%	*Campylobacter* Species
Order Ciconiiformes	85			15	17.6%	
White stork *(Ciconia ciconia)*	85	O	Y	15	17.6%	*C. jejuni* (14), *C. coli* (1)
Order Columbiformes	28			0	0%	
Collared dove *(Streptopelia decaocto)*	8	H	N	0	0%	
Ringdove *(Columba palumbus)*	20	H	N	0	0%	
Order Passeriformes	24			8	33.3%	
Barn swallow *(Hirundo rustica)*	3	I	N	0	0%	
Common blackbird *(Turdus merula)*	4	O	N	1	25%	*Campylobacter* spp. (1)
Common crow *(Corvus corone)*	2	O	Y	1	50%	*C. jejuni* (1)
Common raven *(Corvus corax)*	1	O	Y	1	100%	*C. lari* (1)
European robin *(Erithacus rubecula)*	1	I	N	0	0%	
House sparrow *(Passer domesticus)*	2	O	N	1	50%	*Campylobacter* spp. (1)
Red-rumped swallow *(Cecropis daurica)*	1	I	N	1	100%	*C. jejuni* (1)
Magpie *(Pica pica)*	5	O	Y	0	0%	
Spotless starling *(Sturnus unicolor)*	2	O	N	1	50%	*C. jejuni* (1)
Western jackdaw *(Coloeus monedula)*	3	O	Y	2	66.7%	*C. jejuni* (2)
TOTAL	137			23	16.8%	

* Taxonomy according to the Handbook of the Birds of the World and BirdLife International digital checklist of the birds of the world: version 7 (June 2023). ^a^ Main feeding diet was divided into 3 categories: H = herbivore, I = insectivore, O = omnivore. ^b^ Presence in landfills was divided into 2 categories: Y = yes, N = no. In the *Campylobacter* Species column, the number of isolates from each species are in brackets ().

**Table 3 vetsci-11-00210-t003:** Factors associated with *Campylobacter* spp. presence in urban birds.

Variable	Category	No. of Samples	Presence	(%)	Chi-Square/Fisher Exact Test	Univariate Logistic Regression			Multivariate Logistic Regression		
					*p*-Value	*p*-Value	OR	CI_95%_	*p*-Value	OR	CI_95%_
Order	Ciconiiformes	85	15	17.6%	0.006		Reference			Reference	
	Columbiformes	28	0	0%		-	-	-	-	-	-
	Passeriformes	24	8	33.3%		0.102	2.33	0.54–6.44	0.074	35.37	0.70–1779.96
Feeding	Herbivore	28	0	0%	0.029	-	-	-	-	-	-
	Insectivore	5	1	20%			Reference				
	Omnivore	104	22	21.1%		0.951	1.07	0.11–10.09	-	-	-
Presence in landfill	Yes	96	19	19.8%	0.213		Reference				
	No	41	4	9.7%		0.159	2.28	0.72–7.19	-	-	-
Season of sampling	Winter	19	3	15.8%	0.004	0.16	3.31	0.61–18.04	0.12	5.05	0.66–38.86
	Spring	56	3	5.3%			Reference			Reference	
	Summer	47	15	31.9%		0.002	8.28	2.22–30.84	0.029	8.78	1.31–58.74
	Autumn	15	2	13.3%		0.3	2.72	0.41–17.98	0.15	5.98	0.6–59.93
Age	Adult	82	8	9.8%	0.010		Reference			Reference	
	Young	55	15	27.3%		0.01	3.47	1.35–8.88	0.049	7.53	1.01–56.35

The dependent variable is the *Campylobacter* spp. presence/absence status. OR: Odds ratio, CI_95%_: 95% Confidence Interval.

**Table 4 vetsci-11-00210-t004:** Results of antimicrobial susceptibility testing (AST) of *Campylobacter* isolates from urban birds.

Antimicrobials (Range, mg/mL)	*Campylobacter* Species	MIC (>mg/L)	No. of Isolates at Each MIC (mg/mL)											Resistant Isolates (%)	
			0.12	0.25	0.5	1	2	4	8	16	32	64	128	Among *Campylobacter* Species	Among Total of *Campylobacter* Isolates
Ciprofloxacin (0.12–16)	*C. jejuni*		4 *				*2*	*8*	*1*					11/15 (73.3%)	12/17 (70.6%)
	*C. coli*	0.5	1 *											0/1 (0%)	
	*C. lari*						*1*							1/1 (100%)	
Nalidixic acid (1–64)	*C. jejuni*	16.0				1 *	3			2	*3*	*6 ^*		9/15 (60%)	9/17 (52.9%)
	*C. coli*						1							0/1 (0%)	
	*C. lari*								1					0/1 (0%)	
Tetracycline (0.5–64)	*C. jejuni*	1.0			5 *					*2*	*6*	*2 ^*		10/15 (66.7%)	11/17 (64.7%)
	*C. coli*	2.0										*1 ^*		1/1 (100%)	
	*C. lari*	1.0												0/1 (0%)	
Streptomycin (0.25–16)	*C. jejuni*	4.0			11	3	1							0/15 (0%)	1/17 (5.8%)
	*C. coli*									*1 ^*				1/1 (100%)	
	*C. lari*					1								0/1 (0%)	
Gentamycin (0.12–16)	*C. jejuni*	2.0	8 *	6										0/15 (0%)	0/17 (0%)
	*C. coli*				1									0/1 (0%)	
	*C. lari*			1										0/1 (0%)	
Erythromycin (1–128)	*C. jejuni*	4.0				15 *								0/15 (0%)	0/17 (0%)
	*C. coli*	8.0				1 *								0/1 (0%)	
	*C. lari*	4.0				1 *								0/1 (0%)	

Isolates categorized as resistant are shown in *italics*. Dark-grey cells are concentrations not available in the Sensititre^®^ EUCAMP2 plate. Light-grey cells are considered resistant. * Highly susceptible isolates: no growth in the lowest concentration of the antimicrobial in the Sensititre plate. ^ Highly resistant isolates: growth in the highest concentration of the antimicrobial in the Sensititre plate.

**Table 5 vetsci-11-00210-t005:** Antimicrobial resistance profiles of *Campylobacter* isolates from urban birds.

Antimicrobial Resistance Pattern	Resistant Isolates (n = 14)	*C. jejuni* (%)	*C. coli* (%)	*C. lari* (%)	Avian Species
CIP	2 (14.3%)	1 (8.3%)	-	-	White stork *(Ciconia ciconia)*, 1
		-	-	1 (100%)	Common raven *(Corvus corax)*, 1
TET	1 (7.1%)	1 (8.3%)	-	-	Western jackdaw *(Coloeus monedula)*, 1
CIP TET	1 (7.1%)	1 (8.3%)	-	-	White stork *(Ciconia ciconia)*, 1
CIP NAL	1 (7.1%)	1 (8.3%)	-	-	White stork *(Ciconia ciconia)*, 1
CIP NAL TET	8 (57.1%)	8 (66.7%)	-	-	White stork *(Ciconia ciconia)*, 8
TET STR	1 (7.1%)	-	1 (100%)	-	White stork *(Ciconia ciconia)*, 1

Percentages were estimated upon the total resistant isolates, *C. jejuni* n = 12, *C. coli* n = 1, and *C. lari* n = 1. NAL: nalidixic acid, CIP: ciprofloxacin, TET: tetracycline, and STR: streptomycin.

## Data Availability

The data that support the findings of this study are available upon request from the corresponding author (A.M.-G.).

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
