# Peer review of "Exploring the Prevalence and Resistance of Campylobacter in Urban Bird Populations"

_vetsci, 2024, doi:10.3390/vetsci11050210_

Round 1

Reviewer 1 Report

Comments and Suggestions for Authors

This paper entitled “City wings : exploring the prevalence and resistance of Campylobacter in urban bird populations” is very pleasant to read, very detailed, well written and structured. It provides interesting information’s on Campylobacter carriage by urban birds.

Comments :

Question : I don't understand the term 'City wings' in the title. Is this really necessary?

Question: when you write “urban species”, does this mean that these birds only live in an urban environment and are not found in the countryside? Or that these birds can migrate between town and country? This point has to be clarified in the paper?

Line 113 : it's not clear that the cloacal swab was taken as soon as the bird arrived at GREFA. Because, if not, it's also possible that the birds came into contact with other birds at the center and may have been contaminated by others. Clarify this,

Line 135 : I'm surprised you do Gram staining for Campylobacter, it's not necessary, and it is not in ISO 10272. You use biochemical tests to confirm the Campylobacter genus. What are these tests? Describe them or mention them if they are from ISO 10272.

Line 137 : specify FBP

Line 145 : you use the EUCAST ECOFFs, but which year did you use as a reference to choose these cut-offs? In no case do you specify whether you took into account the year of isolation of the strains to choose the right cut-off? Did the cut-offs change over the 5 years of isolation? Please specify,

 Line 147 : add the word (EUCAST) after European Committee on Antimicrobial Susceptibility Testing

Line 172 : change into “Overall, Campylobacter were recovered from 8 to 13 urban ……. . A total of 23 Campylobacter isolates were recovered with C. jejuni the most frequent species…..”

Table 3 : I don't understand the last line of the table where it is written “Susceptible    Resistant”. How does it work?

Line 222 : please specify “…. tend to have higher rates of resistant-Campylobacter (p=0.052) while not significantly different.”

Line 224 : it is written EUVCAMP; I think it is EUCAMP.

Figure 3 : I count 13 isolates and not 14 isolates in the figure. One isolate is missing, the one with CIP-TET pattern.

References :

Line 486  : Campylobacter jejuni and C. coli have to be in italic

Line 500 : Campylobacter jejuni has to be in italic

Line 505 : Campylobacter and Salmonella have to be in italic

Line 516 and line 518 : Campylobacter jejuni has to be in italic

Author Response

Dear Reviewer,

We appreciate your time and comments, that are so important to us for improving our manuscript. You can find the complete response letter attached. Thank you for your suggestions that have improved our manuscript.

Best regards,

Aida Mencía-Gutiérrez.

Reviewer 2 Report

Comments and Suggestions for Authors

This is an interesting article which increases the knowledge on the carriage of pathogens and drug resistant pathogens in wild animals across the world, This is a very under researched area, and it is nice to see another paper which begins to fill a part of the knowledge gap.

The study is well executed and generally well written, and as such I only have minor comments below. I have also tried to offer a suggestion to alter any grammatical issues for the authors, which they can take or modify as they see fit.

Line 25, and 257- Is Campylobacters a word? I wonder if just Campylobacter isolates maybe better?

Line 33- perhaps over the last few years, or last decade may sound better here?

Line 37- I struggled with the ‘resulting individual rate of 16.8%’. Would perhaps something like. ‘with 16.8% of individuals showing positive’ sound better?

Line 38- was significantly higher  in the order …(reword)

Line 74- including Antarctica, with C, jejuni, C. lari and C. coli being the most commonly reported species. (reword) .

Line 99-100- I am not so sure what you are saying here but please reword. Perhaps something like, Due to their increasingly close contact with humans, wildlife may act as a reservoir for zoonotic infections?

Line 111- following a standard protocol…. (reword)

Line 143- is it possible to include the concentrations here just for ease so the reader doesn’t have to go back to the EUCAST tables?

Line 173- im not sure the % here words. 8 of 13 is fine, but not all within a species were positive?

Line 218- three isolated were pansusceptible .,… (reword)

Line 286- However, it is well known that …. (reword)

Line 330- exposure rather than exposition?

Line 344- have also bee described. …(Reword)

Line 247- fluoroquinolone resistant and tetracycline resistant…. (reword)

But overall, an interesting and well carried out and reported study so my congratulations to the authors.

Author Response

(The authors gave the same response as above.)

Reviewer 3 Report

Comments and Suggestions for Authors

The article is interesting considering the importance of Campylobacter on public health since it is the most common bacterial cause of gastroenteritis in the world and knowing the possible sources of transmission can help control it.

However, before publishing this manuscript, the authors should consider the following suggestions:

L.103-106. ...rescue center, as well as identify strains at the species level and determine their susceptibility to antimicrobials. This is a suggestion since it seems like a sentence that refers to the results obtained.

L.128. Please be more descriptive in the methodology. Include separately identification of Campylobacter species by PCR, consider details.

L.130. ISO 130 10272–1:2017 procedures.

L.140. If the authors consider the suggested change at the end of the introduction to be appropriate then they should include the meaning of AST, please.

L.147. Although the reference used to carry out the antimicrobial susceptibility studies is indicated, it is important to indicate how the studies were carried out, that is, was the bacteria in the logarithmic phase of its growth kinetics? How long was the exposure time? etc..

Author Response

(The authors gave the same response as above.)

Round 2

Reviewer 3 Report

Comments and Suggestions for Authors

Check throughout the manuscript that "Gram" is in capital letters

L.134. Gram

Although the reference used to carry out the antimicrobial susceptibility studies is indicated, it is important to indicate how the studies were carried out, that is, was the bacteria in the logarithmic phase of its growth kinetics? How long was the exposure time? etc. –

Authors response: We agree with the Reviewer that those data are very interesting. However, standardized AST procedures do not include the analysis of growth kinetics, so it was not performed with these strains. Nevertheless, we believe it is something very interesting to consider for future research.

Reviewer response: Knowing the growth phase in which a bacteria is found is essential to not overestimate results. Although the standardized AST procedures do not include the analysis of growth kinetics, it was important that the authors consider it since the results authors showed are not reliable. Generally, antimicrobial activity studies are carried out in the exponential phase.